# Heavy Metals in Sediments of Urban Streams: Contamination and Health Risk Assessment of Influencing Factors

**Ewa Wojciechowska** [1], **Nicole Nawrot** [1,*] **, Jolanta Walkusz-Miotk** [2], **Karolina Matej-Łukowicz** [1] **and Ksenia Pazdro** [2]

[1] Faculty of Civil and Environmental Engineering, Gdansk University of Technology, Narutowicza 11/12, 80233 Gdańsk, Poland; esien@pg.edu.pl (E.W.); karluko1@pg.edu.pl (K.M.-Ł.)

[2] Marine Geotoxicology Laboratory, Institute of Oceanology Polish Academy of Sciences, Powstańców Warszawy 55, 81-712 Sopot, Poland; miotk@iopan.pl (J.W.-M.); pazdro@iopan.pl (K.P.)

\* Correspondence: nicnawro@pg.edu.pl

**Abstract:** Sediments of two urban streams in northern Poland outflowing to the Baltic Sea were assessed to explain the spatial variation in relation to urbanization level of the catchment, the role of retention tanks (RTs) and identification of pollution level. During the 3 month period of investigation sediment samples were collected from the inflow (IN) and outflow (OUT) of six RTs located on streams for flood protection. Six heavy metals (HMs) were investigated: Cu, Pb, Zn, Cd, Ni, Cr. The assessment of four geochemical enrichment indices used to quantify contamination of HMs in the sediments at IN and OUT samples was carried out. Contamination factor (CF), pollution load index (PLI), geoaccumulation index ($I_{geo}$) and potential ecological risk (RI) were calculated and the indices usefulness was assessed. Also, the hazard quotient (HQ) was calculated to assess health risk associated with dredging works. In sediments from RTs where paved surfaces constituted more than 70% of the catchment the HMs concentrations were from one to three times higher for Ni and from two to 143 times higher for Cu in comparison to soft catchment results. The extremely high Cu concentration (1114 mg/kg d.w.) found in sediments at RT Orłowska IN was most likely associated with large area of roofs covered with copper sheet. Calculation of CF, PLI, $I_{geo}$, RI, HQ indicators allows for a complex and multi-dimensional assessment of sediment status. Among these, CF and PLI classified the analyzed sediments as most polluted. Basing on the sedimentary HMs concentrations the health risk level via dermal exposure pathway was assessed as low.

**Keywords:** heavy metal pollution; retention tanks; urbanization; soft vs paved surfaces; pollution indices

## 1. Introduction

Heavy metals (HMs) are regarded as an important contaminant of the environment, if present in amounts exceeding natural concentrations [1,2]. HMs are a natural component of rocks. As the result of rocks weathering they are transferred to soil and bottom sediments, where they are supplemented with metals originating from anthropogenic activity such as urbanization, industrialization, transportation and energy production [3,4]. One important yet relatively seldom studied means of metals transferring to sediments is storm water run-off. Contemporary studies [2,5–10] revealed that storm water run-off carried metals originating from a variety of everyday activities associated with tire wear, corrosion, roof run-off and fuel combustion products. It is well established that run-off from urbanized catchment is abundant in HMs [11–13]. However, little is known as regards the influence of the run-off from urbanized catchment composed of soft, natural surfaces (forest, lawns, gardens) vs run-off from

paved surfaces (roads, sidewalks, squares, fuel stations) on the metals levels in the run-off receivers. The types of pavements are not equivalent from the spreading pollutants in watershed management perspective [6]. Within a city different kinds of retention tanks (RTs) are built in order to intercept the overflow of storm water [14,15]. The usefulness of RTs is assessed basing on the volume of storm water that can be retained in the RTs, as it limits the risk of flood. However, due to the flow rate decrease, sedimentation of solids is enhanced in the RTs. This predominantly concerns suspended matter of smaller diameter that is dragged along streams. It is characteristic of the run-off that suspended matter it carries comprises elevated concentration of HMs and that the ratios between the concentrations of individual metals differ in the wide range [16,17]. Zinc, lead, copper, cadmium, nickel and chromium are typically observed in urban catchments sediments [2,5,18]. HMs due to their toxicity, persistence and potential to bioaccumulate present a serious problem of environmental pollution [19–21]. However, little is known as regards the influence of possible health risk due to HMs contamination in bottom sediments. Inhabitants of urban areas could easily be exposed to metals in the environment via three pathways: direct inhalation, ingestion and dermal contact absorption [22–24]. The human exposure to hazardous elements contained in sediments, soils, drinking water etc. is most often calculated as a component of these three pathways. HMs occurrence can be especially detrimental to the children health due to weakness of their immune system [25]. The particularly dangerous effects of heavy metals have been extensively reported for cardiovascular system [26,27], renal system [28,29] and hepatic system [30,31].

　　　Toxicity of a given element is strongly dependent on the element mobility, as soluble species are strongly bioavailable while these absorbed by or adsorbed to are less available. The mobility strongly depends on physico-chemical condition, e.g., metal species that are adsorbed to mineral matrix under neutral pH will likely desorb under acidic conditions of the digestive track and be absorbed by an organism as soluble species. To assess the level of contamination and the environmental and health risks that originate from the metals occurrence, a number of indices is used that indicate the enrichment of a given environmental component as compared to natural concentration of this component. The level of enrichment factor indicates the potential of sediments to show toxic effects to biota under favourable conditions as the actual toxicity to biota of the metals will depend on the physical, chemical and biological conditions under which organisms get in contact with sediments containing the metals. The indices used to describe heavy metal enrichment of sediments include contamination factor (CF), pollution load index (PLI), geoaccumulation index (Igeo), potential ecological risk (RI) and hazard quotient (HQ) [32–35]. Single indices of the above mentioned list are used, as a rule, to characterize sediments contamination. This approach, although enables evaluation of contamination, limits the ability to compare degree of contamination of sediments investigated in different studies. The current study was carried out in Gdańsk, Poland—a city located on the shores of the Baltic Sea. Streams flowing from natural, uncontaminated moraine hills, through highly urbanized residential and industrial areas were investigated. The aims of the study were: (1) assessment of the influence of urban run-off on the concentration of: zinc (Zn), copper (Cu), lead (Pb), cadmium (Cd), chromium (Cr) and nickel (Ni) in sediments of streams receiving run-off from soft and paved catchment; (2) comparison of the five indices commonly used to quantify sediments contamination level with HMs and (3) calculation of hazard quotients caused by exposure of people to sediments contaminated with HMs.

## 2. Materials and Methods

### 2.1. Research Area

　　　The research area is presented in Figure 1. RTs situated along two streams—Strzyża Stream and Oliwski Stream—were investigated. Strzyża Stream is 13.2 km long and it is the longest stream in Gdańsk. The total catchment area is about 3400 ha. The source is located near express road and in its upper part the stream flows through the forests. In its middle part Strzyża flows in an open channel through densely populated city districts, close to busy traffic arteries. The lower part of the riverbed is

channeled. The stream outflows to Dead Vistula branch of the Vistula River 5 km before its discharge to the Baltic Sea. Numerous outlets of the storm water interceptors discharge into the Strzyża Stream as it flows through the urban area. In the current study sediments from three RTs (Figure 1) located in the middle, densely urbanized, part of Strzyża were analyzed.

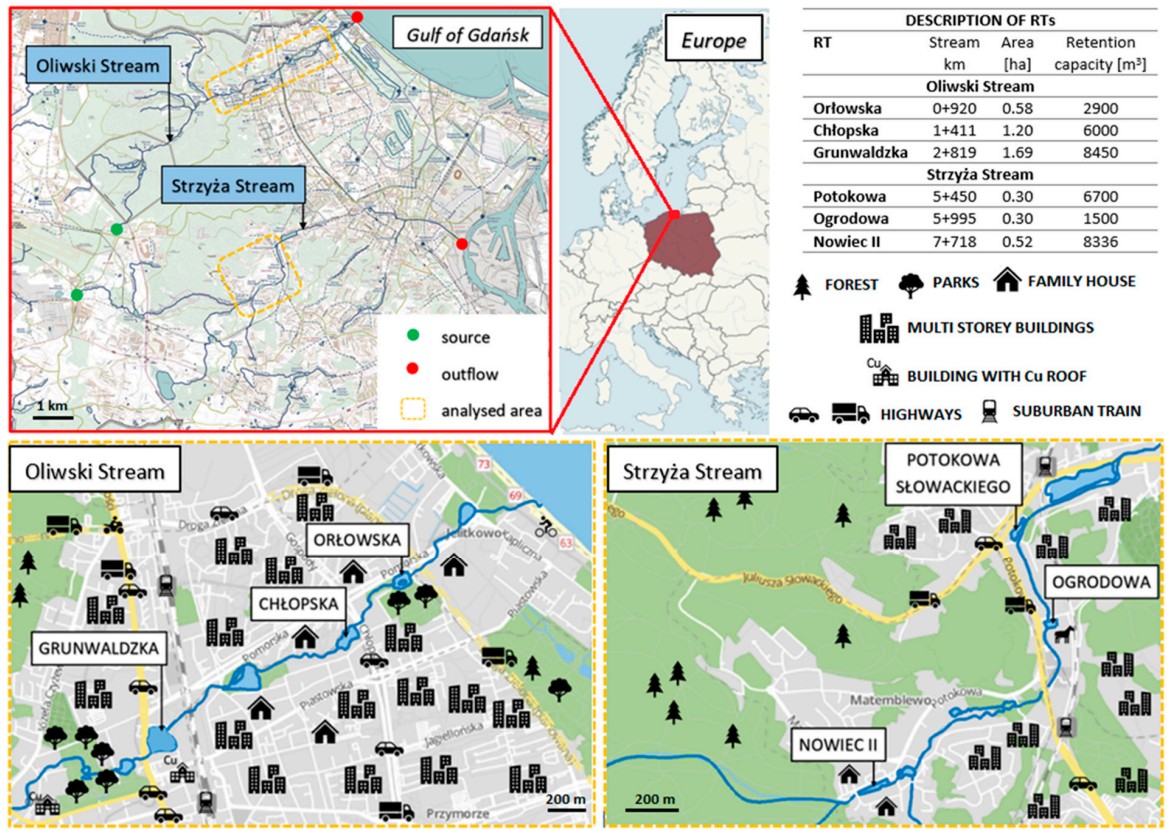

**Figure 1.** Location and basic characterization of analyzed retention tanks on Oliwski Stream and Strzyża Stream.

Oliwski Stream is 9.6 km long and it is the second longest stream in Gdańsk, with a largest average flow rate Q = 158 dm³/s and catchment area of 3050 ha. Its source is located close to the express road. The stream inflows directly to the Gulf of Gdańsk. The upper part of the watershed is covered with forest and green areas with some fish ponds. In the middle and lower part the riverbed runs through densely urbanized residential areas, close to the traffic arteries. Figure 1 presents location and basic characteristics of the three RTs analyzed in the current study.

## 2.2. Sampling

The sediment samples from RTs were collected in the period from March to May 2017 (three series of samples). The samples were collected from three RTs on Strzyża Stream: Nowiec II, Ogrodowa and Potokowa and from three RTs on Oliwski Stream: Grunwaldzka, Chłopska and Orłowska at two points of each tank—near the inflow (IN) and outflow (OUT) (Figure 1). In total 72 sediment samples were collected to investigation. Plexiglas tube pushed into sediments was used to collect the top layer of sediments (with a thickness of approx. 5 cm). On collecting samples were placed in polyethylene bags and within 4 hours delivered to the laboratory.

## 2.3. Soft and Paved Catchments of RTs

In this paper the RTs were divided into two categories: 1st: RT with soft catchment (<70% surface is paved)—only Nowiec II on Strzyża Stream was distinguished here, cause at cities the presence of

natural areas is usually negligible. The 2nd category: RTs retaining stormwater and surface run-off from catchment with paved surface (>70% surface is paved). The 2nd category includes RTs: Ogrodowa, Potokowa, Grunwaldzka, Chłopska and Orłowska.

*2.4. Measurements of HM Concentrations*

　　Collected sediment samples were homogenized and transferred to Petri dishes and lyophilized. Dried sediments sieved through 0.063 mm sieves and sealed in PE bags. Sediment subsample of 0.5 g (0.001 g accuracy) was digested with $HClO_4$, HF and HCl (3:2:1; Suprapur) in teflon bombs in an oven (140 °C) for 4 hrs. Then the solution was evaporated to dryness and 5 ml of concentrated Suprapur $HNO_3$ was added and evaporated. The dried residue was dissolved in 5 mL of Suprapur 0.1 M nitric acid and placed in polyethylene tubes. Dilutions x10, x100 and x1000 were prepared and analyzed in a Flame Atomic Absorption Spectrometer (AAS) using deuterium background correction. The Cd concentrations were measured in inductively coupled plasma mass spectrometry (ICP-MS). The results are presented in mg/kg d.w. The measurements were carried out in three replications. Quality control was assured by analysing certified reference sediments (IAEA-433 and JMS-1 and "blanks", according to the same procedure). Recoveries in the range of 92–103%, depending on individual metals, were achieved. The precision, given as Relative Standard Deviation, was in the range 3–5%. The detection limits (LOD) of each element was calculated as Blank + 3·SD, where SD values were the standard deviations of the blank samples (n = 5). LODs were as follows: Pb = 1.0 µg/g, Zn = 0.5 µg/g, Cr = 1.5 µg/g, Ni = 0.7 µg/g, Cu = 0.3 µg/g, Cd = 35 ng/g (0.035 mg/g).

*2.5. Evaluation of Sediments Pollution by HM*

　　The assessment of HM contamination was based upon German Länder-Arbeitsgemeinschaft Wasser classification (LAWA) [36] consisting of seven classes, according to HM concentrations. The boundary values for each class are described in Table S1.

　　Pollution degree was also assessed using contamination factor (*CF*) [32,33,37], pollution load index (*PLI*) [32–34,37], geoaccumulation index ($I_{geo}$) [38–41], potential ecological risk (*RI*) [23,32,34,37,38] and hazard quotient (*HQ*) [22,23,35,37,38,42].

　　The contamination factor (*CF*) was calculated as (1):

$$CF = \frac{C_{mSample}}{C_{mBackground}} \tag{1}$$

where $C_{mSample}$ is HM in the analysed sample and $C_{mBackground}$ is the geochemical background concentration. According to Hakanson [32], 4 contamination categories are distinguished: *CF* < 1: low contamination, $1 \leq CF < 3$: moderate contamination, $3 \leq CF < 6$: considerable and $CF \geq 6$: very high contamination.

　　The pollution load index (*PLI*) was calculated using Equation (2):

$$PLI = (CF_1 \times CF_2 \times CF_3 \times \ldots \times CF_N)^{1/N} \tag{2}$$

where *CF* is the contamination factor described above and *N* is the number of metals studied. *PLI* < 1 corresponds to perfect sediment quality, *PLI* = 1 shows that only baseline levels of pollutants are present, while *PLI* > 1 indicates deterioration of site quality [33].

　　The index of geoaccumulation ($I_{geo}$) enables assessment of the contamination by comparing the current levels of HM concentrations to the previous status of the research site. The $I_{geo}$ is calculated using Equation (3):

$$I_{geo} = \log_2 \left[ \frac{C_{mSample}}{1.5 \times C_{mBackground}} \right] \tag{3}$$

where $C_{mSample}$ and $C_{mBackground}$ in (3) are as described above. Basing on the $I_{geo}$ value, 7 descriptive classes of sediments contamination presented in Table S2 are defined [37–41].

In this study, the $C_{mBackground}$ value was defined on the basis of average concentration of the analysed metals in the top layer of soil in the Gdańsk urban area [43] (research developed from 1991)—hereinafter referred as geochemical background—GB. The minimum, maximum and mean values are presented in Table S3. The mean values were used for calculations.

To assess the degree of pollution with a given HM the potential ecological risk (*RI*) was also calculated according to formula (4) published by Hakanson [32]:

$$RI = \sum_{i=1}^{n} E_j^i = \sum_{i=1}^{n} T_n^i \cdot \frac{C_{mSample}^i}{C_{mBackground}^i} \tag{4}$$

where $C_{mSample}$ and $C_{mBackground}$ in (4) are as described above and $T_n{}^i$ is the biological toxic factor for HM subscript. The toxic response factors for Cr, Ni, Cu, Cd, Pb and Zn are 2, 6, 5, 30, 5 and 1, respectively [32]. Five degrees of ecological risk for single element $E_j^i$ and four classes of ecological risk *RI* for all factors combined are distinguished (Table S4). Although RI index was originally used as a diagnostic tool for the purpose of controlling water pollution, it was also used for assessing the quality of sediments in terms of HMs pollution [23,32,34,37,38].

Hazard quotient *HQ* [22,23,35,37,38,42] is used to evaluate the health risk to humans related to sediments dredging from RTs. This kind of evaluation is used to quantify both carcinogenic and non-carcinogenic risks to humans via three exposure pathways: ingestion, dermal contact and inhalation. The daily doses via dermal contact ($ADD_{derm}$) were calculated for the employees responsible for dredging of bottom sediments from RTs in Gdańsk. The exposure factors and values used to calculate the intake value and risk are given in Table 1.

**Table 1.** The assumptions used to calculate the average daily doses of potentially toxic metals via dermal contact for employees dredging the sediments from retention tanks in Gdańsk.

| Factor | Definition | Unit | Adults |
|--------|-----------|------|--------|
| EF | Exposure frequency | days/years | 30 |
| ED | Exposure duration | years | 40 |
| BW | Body weight of the exposed individual | kg | 70 |
| AT | Average time of exposure | days | 40 × 365 = 14,600 |
| SA | Exposed skin surface area | cm$^2$ | 4350 |
| AF | Skin adherence factor | mg/(cm·day) | 0.7 |
| ABF | Dermal absorption factor | unitless | 0.001 |

The average daily doses (ADDs) of potentially toxic metals via dermal pathway ($ADD_{derm}$) for adults were estimated using the formula (5) [44–48]:

$$ADD_{derm} = C_m \times \frac{SA \times AF \times ABF \times EF \times ED}{BW \times AT} \times 10^{-6} \left[ \frac{mg}{kg \cdot day} \right] \tag{5}$$

where $C_m$ is the concentration of metal in the analysed medium.

The *HQ* is the ratio of the *ADD* of a HM to its reference dose (*RfD*) for the same exposure pathway(s) [44] and was estimated using Equation (6):

$$HQ = \frac{ADD}{RfD} \tag{6}$$

The sum of *HQ* is defined as hazard index and refers to the total risk of non-carcinogenic element via three exposure pathways. The reference dose (*RfD*) (mg/kg·day) is the maximum daily dose of a metal from a specific exposure pathway, that is believed not to lead to an appreciable risk of deleterious

effects to sensitive individuals during a lifetime [23]. If the *ADD* is lower than the *RfD* − *HQ* ≤ 1, adverse health effects are unlikely to appear, whereas if the *ADD* exceeds the *RfD* − *HQ* > 1, adverse health effects are likely to appear [44,48]. In this study the *HQ* was calculated basing only on the one pathway (dermal contact), due to the assumption of dermal contact of workers with periodically dredged sediments.

### 2.6. Statistical Analyses

The experimental data analysis (mean, standard deviation, maximum and minimum concentrations) using STATISTICA® software (StatSoft Polska, Poland) were performed in this study.

## 3. Results and Discussion

### 3.1. The Watershed Pollution

The average values (±SD) of analyzed HMs are presented in Table 2. The concentrations of 6 analyzed HMs in sediments fluctuated in a wide range, depending on the sampling site and a metal in question. The pH ranges from 6.8 in Potokowa, Ogrodowa and Grunwaldzka RTs to 7.0 in Nowiec II and Chłopska RTs. The contents of organic matter were in the ranges 2–13%.

**Table 2.** The average concentration of HMs (±SD) in bottom sediments from retention tanks on Strzyża Stream and Oliwski Stream [mg/kg d.w.].

| Retention Tank | Sampling Site | Average Concentration of HM in Sediments [mg/kg d.w.] | | | | | |
|---|---|---|---|---|---|---|---|
| | | Cu | Zn | Pb | Cd | Cr | Ni |
| Strzyża Stream | | | | | | | |
| Nowiec II | IN | 3.09 ± 0.05 | 18.2 ± 0.3 | 5.85 ± 0.08 | 0.088 ± 0.000 | 2.50 ± 0.07 | 3.81 ± 0.05 |
| | OUT | 7.80 ± 0.10 | 49.9 ± 0.8 | 13.2 ± 0.1 | 0.111 ± 0.000 | 3.80 ± 0.07 | 4.13 ± 0.05 |
| Ogrodowa | IN | 18.5 ± 0.3 | 95.1 ± 1.5 | 16.7 ± 0.2 | 0.281 ± 0.001 | 5.11 ± 0.10 | 5.42 ± 0.1 |
| | OUT | 37.4 ± 0.6 | 186 ± 3 | 43.2 ± 0.5 | **0.352 ± 0.001** | 8.98 ± 0.2 | 5.98 ± 0.1 |
| Potokowa | IN | **211 ± 3** | **210 ± 3** | **108 ± 1** | 0.150 ± 0.001 | 9.11 ± 0.2 | 6.54 ± 0.10 |
| | OUT | 93.5 ± 1.5 | **356 ± 6** | 49.7 ± 0.6 | **0.60 ± 0.01** | 14.2 ± 0.2 | 10.93 ± 0.10 |
| Oliwski Stream | | | | | | | |
| Grunwaldzka | IN | 37.8 ± 0.6 | 174 ± 3 | 46.1 ± 0.7 | 0.091 ± 0.001 | 12.9 ± 0.2 | 7.13 ± 0.12 |
| | OUT | 50.1 ± 0.8 | **244 ± 4** | 81.9 ± 1.3 | **0.469 ± 0.007** | 14.6 ± 0.2 | 10.3 ± 0.2 |
| Chłopska | IN | 55.1 ± 0.9 | 45.0 ± 0.8 | 22.7 ± 0.3 | 0.300 ± 0.008 | 17.4 ± 0.4 | 3.80 ± 0.06 |
| | OUT | 64.9 ± 1.1 | 103 ± 2 | 35.8 ± 0.6 | 0.102 ± 0.002 | 25.5 ± 0.3 | 4.25 ± 0.07 |
| Orłowska | IN | **1114 ± 19** | 136 ± 2 | **160 ± 2** | 0.092 ± 0.001 | 12.8 ± 0.2 | 6.27 ± 0.11 |
| | OUT | 40.9 ± 0.7 | 82.1 ± 1.4 | 29.1 ± 0.4 | 0.200 ± 0.000 | 14.6 ± 0.2 | 6.66 ± 0.09 |

Note: **Bold**—elevated/outstanding concentrations of heavy metals.

HMs distribution in sediments of Strzyża Stream: The sediments from Nowiec II were characterized by very low concentrations of analysed HMs, much lower than the GB concentrations for surrounding soils (Table S3). This RT is located right below the forested part of watershed and belong to the 1st category (soft catchment). Concentrations of Zn, Cd, Cr and Ni in sediments tend to increase in the subsequent RTs along Strzyża reflecting the rapidly growing urbanization (e.g., population density) of the area and suggesting anthropogenic origin of these HMs. In the case of Cu and Pb at Strzyża Stream the spatial variations are similar, while the highest concentrations of Cu and Pb were observed at the IN site in the Potokowa RT. Two storm water drainage system outlets are located in close proximity to this point. Urban rapid transit rail (fueled by diesel oil) passing directly next to the Potokowa RT may also affect the quality of sediments there. The LAWA classification (Table S1) indicates that sediments from Potokowa RT are moderately to significantly polluted in terms of Pb, Cu and Zn concentrations. The potential source of these HMs can be one of the major communication arteries with intense traffic (two lanes in each direction) located at a very close distance to the Strzyża bed below Nowiec II (Figure 1). Zn is used for coating steel to protect against corrosion in cars. Zn could also originate from burning coal in the older residential houses scattered there as

well as from abrasion of car tires. It is worth noting that Zn concentration exceeded II class of LAWA only for the Potokowa IN site. This means that only sediments from Potokowa were contaminated with Zn from external pathways. Pb in the environment is often associated with the use of gasoline products. The unleaded petrol, which is currently in use, still contains Pb, though below 5 mgPb/L. Between the Ogrodowa and Potokowa tanks there is a gardening farm, another one is located a short distance upstream, before Ogrodowa tank. This could be of importance since Cu is a widely used component of insecticides and fungicides. According to LAWA classification Cd, Cr and Ni concentrations in bottom sediments from RTs on Strzyża Stream are low and correspond to unpolluted category with very small anthropogenic interference. The distribution of HMs pollution in the Strzyża Stream is characteristic; concentrations are increasing along the stream as it flows from the green areas to more and more intensively developed city districts. This is most likely due to HMs being associated with the smallest fraction of the suspended solids, which is in agreement with increasing HMs concentrations in the downstream tanks. As sedimentation of larger fraction of suspended solids occurs in the upper RTs, the smallest fraction settles in the downstream located tanks. In comparison to research conducted by Nowell et al. [48] with regard to contamination of stream bottom sediments in relation to urbanization in the USA, lower concentrations of HMs were observed in the current study for all HMs with exception of Pb, Zn and Cu in Potokowa RT. The concentrations of Pb and Zn were on similar level in our study and in the study by Nowell et al. [48] while the Cu concentration was definitely higher in Potokowa tank. The mean values observed in Nowell et al. study were as follows: Cu: 80, Zn: 300, Pb: 100, Cd: 0.8, Cr: 80 and Ni: 50 [mg/kg d.w.].

HMs distribution in sediments of Oliwski Stream: The HMs concentrations were much more diversified than those in the Strzyża Stream sediments. There were no obvious trends in distribution of HMs along the Oliwski Stream. The landscape of the watershed around analysed RTs on Oliwski Stream is diverse; the settlements with multi storey buildings and a substantial impervious part of the area (pavements, intersections, roads) can be distinguished. Elevated concentrations of Zn in comparison to the GB were found in Grunwaldzka and Orłowska RTs, which indicates that anthropogenic activity affects the stream along its entire length; for example, according to LAWA, due to Zn concentration at the Grunwaldzka OUT site. Cu concentrations also exceeded the GB in all sampling points. The last of these locations, Orłowska tank situated close to the Oliwski Stream estuary, was characterized by extremely high Cu concentration reaching 1114 mg/kg d.w. (at the IN site). The Cu concentration at Orłowska IN was clearly higher than those reported in literature—in the range from 16 times more (sediments in open reservoir in Kielce, Poland [49] to 166 times more than sediments in retention pond in Puławy, Poland [17]. In the study conducted by Nowell et al. [48] not only the mean HM concentrations for USA bottom sediments but also the highest values were reported. The highest value of Cu concentrations in the USA was equal to 500 mg/kg d.w. (in Boston). When comparing this report to our study it is obvious that the Cu concentration measured at Orłowska IN (1114 mg/kg d.w.) is over two times as high. Increased concentrations of Cu can be associated with the presence of numerous copper roofs in the catchment area there [50]. Large density of such roofs is located nearby Grunwaldzka RT. Migration of Cu downstream could take place, for example after urban flash floods [51]. High spatial variability of HMs concentrations in bottom sediments before and after the flood in July 2016 was suggested [52,53]. The Cu concentration at the Ogrodowa IN sampling point before flood was 30 times lower, while at Grunwaldzka tank three times higher than concentrations in sediments collected after the flood. Another source of Cu in the watershed could be intensively fertilized gardens. The GB concentration of Pb (39 mg/kg d.w.) was also exceeded in several locations, including Grunwaldzka RT (sites IN, OUT) and Orłowska (site OUT). The average values of the GB for Cr (8.95 mg/kg) and Ni (5.65 mg/kg) were exceeded 1.3–2.8 and 1.1–1.8 times respectively, however according to LAWA the sites were classified as unpolluted. With reference to Cu, Zn and Pb LAWA classifies selected bottom sediments as moderately to ultimately polluted (Orłowska IN site).

In comparison to HMs concentrations in the 2nd RT category (>70% paved) the important differences were distinguished. The HMs concentrations in sediments from the 2nd category RTs were higher for all analysed metals than in Nowiec II RT (1st category): for Cu from two to 143, for Zn from three to seven, for Pb from three to 12, for Cd from one to five, for Cr from two to seven and for Ni from one to three times higher.

### 3.2. Evaluation of Sediments Pollution Using Indices

Results of CF, $I_{geo}$ and PLI calculated using GB for sediments from Strzyża and Oliwski Streams are shown in Table 3.

**Table 3.** Contamination factor (CF), geoaccumulation index ($I_{geo}$) and pollution load index (PLI) for bottom sediments collected from Strzyża and Oliwski Streams.

| Retention Tank | Point | Cu | | Zn | | Pb | | Cd | | Cr | | Ni | | PLI |
|---|---|---|---|---|---|---|---|---|---|---|---|---|---|---|
| | | CF | $I_{geo}$ | CF | $I_{geo}$ | CF | $I_{geo}$ | CF | $I_{geo}$ | CF | $I_{geo}$ | CF | $I_{geo}$ | |
| | | | | | | Strzyża Stream | | | | | | | | |
| Nowiec II | IN | 0.7 | −1.0 | 0.4 | −1.9 | 0.4 | −2.1 | 0.4 | −2.1 | 0.5 | −1.5 | 0.7 | −1.2 | 0.5 |
| | OUT | 1.9 | 0.3 | 1.1 | −0.4 | 0.8 | −0.9 | 0.4 | −1.8 | 0.8 | −0.9 | 0.7 | −1.0 | 0.9 |
| Ogrodowa | IN | 4.4 | 1.6 | 2.1 | 0.5 | 1.0 | −0.6 | 1.1 | −0.4 | 1.1 | −0.5 | 1.0 | −0.7 | **1.5** |
| | OUT | **8.9** | 2.6 | 4.1 | 1.5 | 2.6 | 0.8 | 1.4 | −0.1 | 1.9 | 0.3 | 1.0 | −0.5 | **2.5** |
| Potokowa | IN | 50 | **5.1** | 4.7 | 1.6 | **6.6** | 2.1 | 0.6 | −1.3 | 1.9 | 0.4 | 1.2 | −0.4 | **3.6** |
| | OUT | 22 | 3.9 | **7.9** | 2.4 | 3.0 | 1.0 | 2.4 | 0.7 | **3.0** | 1.0 | 1.9 | 0.4 | **4.4** |
| | | | | | | Oliwski Stream | | | | | | | | |
| Grunwaldzka | IN | 2.2 | 0.6 | 1.4 | −0.1 | 1.2 | −0.3 | 0.1 | −3.9 | 1.4 | −0.1 | 1.3 | −0.2 | 0.9 |
| | OUT | 3.0 | 1.0 | 2.0 | 0.4 | 2.1 | 0.5 | 0.5 | −1.5 | 1.6 | 0.1 | 1.8 | 0.3 | **1.6** |
| Chłopska | IN | 3.3 | 1.1 | 0.4 | −2.0 | 0.6 | −1.4 | 0.3 | −2.1 | 1.9 | 0.4 | 0.7 | −1.2 | 0.8 |
| | OUT | 3.8 | 1.4 | 0.9 | −0.8 | 0.9 | −0.7 | 0.1 | −3.7 | 2.8 | 0.9 | 0.8 | −1.0 | **1.0** |
| Orłowska | IN | 66 | **5.5** | 1.1 | −0.4 | 4.1 | 1.5 | 0.1 | −3.8 | 1.4 | −0.1 | 1.1 | −0.4 | **1.9** |
| | OUT | 2.4 | 0.7 | 0.7 | −1.1 | 0.7 | −1.0 | 0.2 | −2.7 | 1.6 | 0.1 | 1.2 | −0.3 | 0.9 |

Note: **Bold**—results showing the highest values of pollution.

Strzyża Stream: The sediments from Nowiec II were characterized by CF as slightly contaminated, except OUT where due to Cu and Zn sediments were moderately contaminated. At the same time $I_{geo}$ indicated uncontaminated sediments (0 class), except from Cu in OUT—witch varied from uncontaminated to moderately contaminated (class 1). Ogrodowa and Potokowa sediments were moderately to very highly contaminated, especially due to Cu at OUT in Ogrodowa and at both point in Potokowa. The $I_{geo}$ rating was very similar—the values ranged from class 1 to 5 (extremely contaminated sediments). Olubunmi and Olorunsola [54] assessed pollution of sediments of Agbabu Bitumen Deposit Area (Nigeria) with $I_{geo}$, reporting 0.31 for Pb, 0.13 for Zn, 0.30 for Cu, 0.16 for Cr, –1.49 for Cd and 0.86 for Ni. The $I_{geo}$ values for Cu, Pb, Zn, Cd and Cr in Ogrodowa and Potokowa were higher, while for Ni in all samples on Strzyża Stream were lower than $I_{geo}$ of Agbabu River. The PLI was higher than the reference line (PLI = 1) for Ogrodowa and Potokowa tanks. According to Likuku et al. [41] PLI ≥ 1 indicates the need of immediate intervention to prevent pollution, 0.5 ≤ PLI < 1 suggests that more detailed study is needed to monitor the site, whilst a value below 0.5 indicates that there is no need of intervention. The CF and PLI were used by Likuku et al. [41] to evaluate the soil pollution around the copper-nickel mine in Selebi Phikwe Region. The CF values reported were as follows: Pb − CF = 2.16, Zn − CF = 1.21 and Cu − CF = 6.36, Ni − CF = 3.64. The PLI was 1.63. The corresponding values for RTs on Strzyża Stream in the current study were for Nowiec II and Ogrodowa IN lower than reported by Likuku et al. [41], while for Ogrodowa OUT and Potokowa they were higher. The CF for Cu in Potokowa tank gives definitely high value which is actually beyond the range of values presented by the descriptive scale.

Oliwski Stream: For sediments from Oliwski Stream the worst value of CF was received due to the content of Cu. It can be explained by really high $C_{mSample}$ value in comparison to $C_{mBackground}$ in analyzed sediments. Also, for other RTs CF reached the worst values for Cu. Only the sediments

from Chłopska RT—due to Zn, Pb, Cd and Ni and Orłowska—due to Cd can be considered as slightly contaminated. PLI exceeded 1 for Chłopska OUT and Orłowska IN. The range of PLI in sediments along Nakivubo channelized stream flowing through urban and industrial area was 1.25–2.5 [2], so it was higher than in Oliwski Stream. In most of RTs on Oliwski Stream the maximal $I_{geo}$ values were below 1.5, which corresponded to the moderate contamination of sediments. Only for Orłowska IN the $I_{geo}$ was in class 6. The $I_{geo}$ for Zn exceeded the Agbabu values [54] only at Grunwaldzka OUT (class 1), while the $I_{geo}$ for Cu was higher at all RTs. The $I_{geo}$ for Pb was higher than for Agbabu in two samples—Grunwaldzka OUT (1 class) and Orłowska IN (class 2). Cr was higher in the case of Chłopska RT (1 class). Chen et al. [55] reported the class 5 of $I_{geo}$ in sediments of Kaohsiung Harbor in Taiwan only in reference to Hg, the class 4 for Cd while due to Cu the class 2 level was noted.

## 4. Potential Ecological Risk Factor

The results of ecological risk factor $E_j^i$ and potential ecological risk RI for Strzyża and Oliwski streams are presented in Table 4. The considerable risk was noted for Potokowa (OUT) − $E_j^i$ = 111. High risk was noted for Potokowa (IN) − $E_j^i$ = 252 and Orłowska (IN) − $E_j^i$ = 330 due to Cu concentrations. The RI was high for Potokowa (IN, OUT) − RI = 302 and RI = 218 and considerable for Orłowska (IN) − RI = 363. The highest values are determined not so much by the biological toxic factor as ($T_n^i$ = 5) (Equation (4)) but by the extremely high Cu concentration in the analyzed sediments. The high RI of sediments indicates that the ecosystem services of RTs can be affected and clearly shows that HMs deposited in the sediments pose ecological hazard [19,56]. In case of other RTs in the study area the $E_j^i$ values for Zn, Pb, Cd, Cr, Ni and Cu were between 0 and 72, much lower than the minimum risk threshold (80) [32] resulting in very low probability of combined ecological risk. The monomial ecological risk factor $E_j^i$ indicated that the severity of pollution of the six HMs decreases in the following sequence: Cu > Cd > Pb > Ni > Cr > Zn, highlighting the risk posed by Cu to the ecosystem.

**Table 4.** $E_j^i$ and RI values for Strzyża Stream and Oliwski Stream.

| Retention Tank | Point | $E_j^i$ | | | | | | RI |
|---|---|---|---|---|---|---|---|---|
| | | **Cu** | **Zn** | **Pb** | **Cd** | **Cr** | **Ni** | |
| Strzyża Stream | | | | | | | | |
| Nowiec | IN | 4 | 0 | 2 | 21 | 2 | 4 | 32 |
| | OUT | 9 | 1 | 4 | 13 | 2 | 4 | 34 |
| Ogrodowa | IN | 22 | 2 | 33 | 42 | 2 | 13 | 115 |
| | OUT | 45 | 4 | 5 | 34 | 6 | 12 | 106 |
| Potokowa | IN | **252** | 5 | 13 | 18 | 5 | 9 | **302** |
| | OUT | **111** | 8 | 15 | 72 | 4 | 8 | **218** |
| Oliwski Stream | | | | | | | | |
| Grunwaldzka | IN | 11 | 1 | 6 | 3 | 2 | 11 | 34 |
| | OUT | 15 | 2 | 11 | 16 | 2 | 8 | 53 |
| Chłopska | IN | 16 | 0 | 3 | 10 | 3 | 4 | 37 |
| | OUT | 19 | 1 | 5 | 4 | 2 | 5 | 35 |
| Orłowska | IN | **330** | 1 | 21 | 3 | 2 | 7 | **363** |
| | OUT | 12 | 1 | 4 | 7 | 2 | 7 | 32 |

Note: **Bold**—elevated/outstanding values according to corresponding classes.

Comparison of sediments contamination based on different indices generally shows similar results and trends, for instance increasing contamination of sediments downstream in Strzyża Stream. However, among the used indices, the most restrictive indicator is CF, followed by PLI > $I_{geo}$ > RI > $E_j^i$. PLI indicator gives a general result of the level of sediment pollution, based on CF individual values for specific metals. In some cases when HMs concentrations are at the class boundary, the indices may give a different description of the situation. For example, in case of the Orłowska tank

it was noted that due to Cu concentration the contamination based on CF was very high, $I_{geo}$ was assessed as strongly contaminated, RI was considerable, $E_j^i$ values suggest the risk was high. At the same time LAWA indicated unlimited pollution. It is of interest that PLI was higher for Potokowa RT than for Orłowska RT, though the HMs concentrations in both RTs were similar (apart from Cu—much higher in Orłowska). This is associated, among other, with the geochemical background values for both RTs. It can be concluded that in case of the analyzed research sites CF and PLI are the best and most restrictive tools for assessing pollution with heavy metals.

## 5. Health Risk Assessment Based on Hazard Quotient

Based on the $RfD_{derm}$ and $ADD_{derm}$ the HQ presented in Table 5 was calculated. The highest values of HQ via dermal contact were noted for both streams for Cr: $5.74 \times 10^{-3}$ and $1.18 \times 10^{-2}$ for Strzyża and Oliwski Stream, respectively. The values of this indicator, which refers to the health hazard of HMs to the workers, decreased in the order of Cr > Pb > Cd > Cu > Zn > Ni. All results of HQ basing on dermal contact with sediments were lower than 1, indicating lack of carcinogenic risk for employees with direct contact with this medium for about 40 years working in this position (assumptions for calculations are listed in Table 1). The HQ > 1 would be observed if the average daily doses will be 100 times higher for Cr, and over the 16000 times higher for Ni, or in case of higher exposure frequency (over 30 days/years). With regard to the maximum concentrations in sediments measured in Orłowska tank for Pb: 160 mg/kg d.w. and for Chłopska for Cr: 25.5 mg/kg d.w., assuming an EF (exposure frequency) increase to 365 days a year (other assumptions unchanged), the highest calculated HQ values were 0.0133 and 0.0185 respectively. These values are again more than 50 times lower than HQ = 1. In this calculation ADD via ingestion and inhalation contact were omitted, assuming no consumption and no dust generated during dredging (due to high water content in the sediments). The complex analyses of health hazard would require also evaluation of ingestion and inhalation pathways. The human health risk assessment is a powerful tool for distinguishing the toxic heavy metals and exposure routes of most concerns in urban environments [23]. Further research should track on any increase of Cr concentration in sediments, since (as it has been shown) Cr poses the highest risk to human health.

**Table 5.** The reference doses ($RfD_{derm}$), average daily dosed via dermal ($ADD_{derm}$) and Hazard Quotient for the average concentrations of HM at Strzyża and Oliwski Stream.

| Metals | Cm [mg/kg] | $RfD_{derm}$ [mg/kg·day] | $ADD_{derm}$ [mg/kg·day] | HQ |
|---|---|---|---|---|
| | | Strzyża Stream | | |
| Cu | 61.9 | $1.20 \times 10^{-2}$ | $2.70 \times 10^{-6}$ | $2.25 \times 10^{-4}$ |
| Zn | 152 | $6.00 \times 10^{-2}$ | $6.65 \times 10^{-6}$ | $1.11 \times 10^{-4}$ |
| Pb | 39.6 | $5.25 \times 10^{-4}$ | $1.72 \times 10^{-6}$ | $3.28 \times 10^{-3}$ |
| Cd | 0.281 | $1.00 \times 10^{-5}$ | $1.21 \times 10^{-8}$ | $1.21 \times 10^{-3}$ |
| Cr | 7.91 | $6.00 \times 10^{-5}$ | $3.44 \times 10^{-7}$ | $5.74 \times 10^{-3}$ |
| Ni | 7.97 | $5.40 \times 10^{-3}$ | $3.47 \times 10^{-7}$ | $6.42 \times 10^{-5}$ |
| | | Oliwski Stream | | |
| Cu | 227 | $1.20 \times 10^{-2}$ | $9.88 \times 10^{-6}$ | $8.24 \times 10^{-4}$ |
| Zn | 130 | $6.00 \times 10^{-2}$ | $5.70 \times 10^{-6}$ | $9.50 \times 10^{-5}$ |
| Pb | 62.7 | $5.25 \times 10^{-4}$ | $2.73 \times 10^{-6}$ | $5.20 \times 10^{-3}$ |
| Cd | 0.210 | $1.00 \times 10^{-5}$ | $9.09 \times 10^{-9}$ | $9.09 \times 10^{-4}$ |
| Cr | 16.3 | $6.00 \times 10^{-5}$ | $7.11 \times 10^{-7}$ | $1.18 \times 10^{-2}$ |
| Ni | 6.40 | $5.40 \times 10^{-3}$ | $2.78 \times 10^{-7}$ | $5.16 \times 10^{-5}$ |

## 6. Conclusions

In summary, the sediments in RTs on the Gdańsk streams accumulate HMs. This fact has a positive aspect, because this may limit their discharge to the Gulf of Gdańsk. On the other hand, it is of significance to conduct assessments of metal content and their variability over time as well as careful handling and utilization of sediments. It is important to estimate if and to what extent metals are mobile and whether they can be released to water causing re-contamination and transport downstream and finally reaching the marine environment. It is also valid to assess the origin of metals in order to eliminate at least some of the potential sources. Concentrations of the analyzed HMs show increasing tendency along the Strzyża Stream reflecting growing urbanization of the catchment area. No obvious tendency in metal concentrations from subsequent tanks was found along Oliwski Stream, where anthropogenic activities were spread more evenly

The enrichment occurred for the entire length; the relocation of HMs is possible. CF, PLI, $I_{geo}$, RI, HQ indicators allow for a better assessment of sediment status. The LAWA classification is helpful in the preliminary assessment of the level of pollution, preceding the indices calculation. In case of the analyzed research sites, CF and PLI can be considered as the best tools for assessing pollution with heavy metals since they set the most stringent requirements. The direct health hazard assessment for workers employed for sediment dredging did not indicate the existence of threats - however this issue is worth further monitoring (especially due Cr concentrations).

**Supplementary Materials:** The following are available online at http://www.mdpi.com/2071-1050/11/3/563/s1, Table S1: Classes of bottom sediments quality for Strzyża Stream and Oliwski Stream with reference to LAWA classification [36], Table S2: Denotation of geochemical classification Igeo for bottom sediments [37–41], Table S3: Concentration of HMs in soils of analysed districts of Strzyża Stream and Oliwski Stream - GB [mg/kg d.w.] [43], Table S4: Degrees of ecological risk (Eji) (for single element) [23,32,34,37,38] and ecological risk (RI) for all factors HM contaminated sediments [32].

**Author Contributions:** Conceptualization, E.W. and N.N.; Methodology, N.N. and J.W.-M.; Software, N.N.; Validation, K.P., J.W.-M. and N.N.; Formal Analysis, E.W., N.N. and K.P.; Investigation, N.N., J.W.-M. and K.M.-Ł.; Resources, N.N. and K.M.-Ł.; Data Curation, X.X.; Writing-Original Draft Preparation, E.W. and N.N.; Writing-Review & Editing, E.W., N.N. and K.P.; Visualization, N.N. and K.M.-Ł; Supervision, E.W. and K.P.; Project Administration, E.W.; Funding Acquisition, E.W.

**Funding:** The work was completed under a GRAM grant, awarded in a competitive procedure by the Dean of the Faculty of Civil and Environmental Engineering, Gdansk University of Technology. The grants are funded from science funds as specified in Journal of Laws no. 96, heading 615, as amended.

**Acknowledgments:** The authors wish to thank Janusz Pempkowiak from Institute of Oceanology Polish Academy of Science for his assistance and support throughout this study.

**Conflicts of Interest:** The authors declare no conflict of interest.

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
