# Peer review of "Heavy Metals in Sediments of Urban Streams: Contamination and Health Risk Assessment of Influencing Factors"

_sustainability, doi:10.3390/su11030563_

Reviewer 1 Report

The work is of interest from the point of view of the contamination of water courses in urban areas and the environmental and health risks that the potential accumulation of pollutants can represent. It is not a subject that has been extensively studied. The work is very well written. All sections are balanced in extension. The objectives are clear and the methodology used is adequate. The discussion is timely and uses appropriate bibliography. The conclusions are adjusted to the results obtained. The tables are adequate and have good quality. The general assessment is: Suitable for publication, once typography errors have been corrected that affect lines 144 to 145.

Author Response

Dear Reviewer,

On behalf of my co-authors I would like to thank you for your positive review. We took into account the comments of each of the Reviewers. Some changes in the manuscript were made and marked.

I hope you will find our revised manuscript suitable for publication in Sustainability.

Sincerely yours,

Nicole Nawrot

Reviewer 2 Report

Major comments:

I did have a few comments that the authors should consider as they revise the paper. Therefore, I recommend that a careful revision is warranted. I explain my concerns in more detail below.

I think it is important for readers to show in the article that when choosing six HM within this set there are two groups: First hazard class (Cd, Pb, Zn) and second hazard class (Ni, Cu, Cr). In addition, the enrichment factor of surface waters (normalized by iron) for Cd is 200, and for Cr it is 0.7. In terms of toxicity in the aquatic environment, Cr is inferior to five other HM. That is, it is necessary to clearly show which of the five selected heavy metals these features can take into account?

Specific Comments:

L17 Five indices are named, but four are named on L 18-19. Where is the HQ (L127)?

L18 has no lower case for Igeo. Compare to L127

L24 What is the reason for choosing two indices out of five?

L84 3 - upper case

L96 here and hereinafter - L132 and others… to L409. According to the text one should use a long dash.

L102-103 Other size

L109 space to gram

L119 Up to% space, compare to L104. Unify by Journal requirements.

L121 Comma must be near g (L120)

L125 there is no space: Tab. S.1.

L128 One index was justified by four authors in 1980-2017. What did the later work bring? (see also L158). Link 2017 should be after 2015.

Formula (3) - ... nd clearer

L144-145 Jumbled Text - Fix

L151 The author did not develop, and published in 1980

Formula (4) format

L158 - compare with L 128, there are 4 other authors!

Formula (5) Shire multiplication sign with Formula (2). To unify

L167 Since Formula (5) is earlier than Table 1, it should be above Formula (5).

Formula (6) format

L187 space after ±

L188 Compare [mg / kg] in Table 5 and L188, in Tab.Sup.3. must be given “mg / kg d. w.” in the method - L115 (as only on L188 it is indicated for the first time). If authors have, it is advisable to show pH values.

L 284 Table 3: Potokowa, Cu  (50 (IN) and 22(OUT); Orlowska,  Cu  (66 (IN) and 2.4(OUT); and  Table 4: Cu  (252 (IN) and 111(OUT); Orlowska,  Cu  (330 (IN) and 12(OUT). These unexpected data require a more detailed explanation of the reasons.

L369 It is advisable to specify the date, because resource may be irrelevant

L 373, 375 It is not clear what these sources are? If published, where are the dates? And etc.

L386 See “control.a”?

L447-455 L164, where the US EPA

L454 See “vol I;.”?

L469 572–589.

Table Sup.2 Compare with L142, where there is no this source - Muller, 1969

Table Sup.2 No lower case in the head of the Table - Igeo

Table Sup.4 In the signature upper and lower case in Eji

Table Sup.4 I suggest brackets after the names of the indices (Eji) and (RI)

Once again I ask you to see if there is a contradiction in the conclusions: according to L24 in the abstract and L 345-346 in conclusion.

Author Response

Dear Reviewer,

On behalf of my co-authors I would like to thank you for careful reading of our manuscript and for very precise comments. In PDF file please find the answers to your comments. The changes in the manuscript were also marked.

I hope you will find our revised manuscript suitable for publication in Sustainability.

Sincerely yours,

Nicole Nawrot

Reviewer 3 Report

Thanks for the excellent work you do. My comments can be found below.

Introduction:

After line 36 talk about some of the systems it can affect in the body such as the:

1)      cardiovascular system

Lanphear, Bruce P., Stephen Rauch, Peggy Auinger, Ryan W. Allen, and Richard W. Hornung. "Low-level lead exposure and mortality in US adults: a population-based cohort study." The Lancet Public Health 3, no. 4 (2018): e177-e184.

 Obeng-Gyasi, Emmanuel, Rodrigo X. Armijos, M. Margaret Weigel, Gabriel M. Filippelli, and M. Aaron Sayegh. "Cardiovascular-Related Outcomes in US Adults Exposed to Lead." International journal of environmental research and public health 15, no. 4 (2018): 759.

2)      Renal system

 Harari, Florencia, Gerd Sallsten, Anders Christensson, Marinka Petkovic, Bo Hedblad, Niklas Forsgard, Olle Melander et al. "Blood Lead Levels and Decreased Kidney Function in a Population-Based Cohort." American Journal of Kidney Diseases (2018).

 Lin, Ja-Liang, Dan-Tzu Lin-Tan, Kuang-Hung Hsu, and Chun-Chen Yu. "Environmental lead exposure and progression of chronic renal diseases in patients without diabetes." New England Journal of Medicine 348, no. 4 (2003): 277-286.

3)      Hepatic system

 Obeng-Gyasi, Emmanuel, Rodrigo X. Armijos, M. Margaret Weigel, Gabriel Filippelli, and M. Aaron Sayegh. "Hepatobiliary-Related Outcomes in US Adults Exposed to Lead." Environments 5, no. 4 (2018): 46.

Can, S., C. Bağci, M. Ozaslan, A. I. Bozkurt, B. Cengiz, E. A. Cakmak, R. Kocabaş, E. Karadağ, and M. Tarakçioğlu. "Occupational lead exposure effect on liver functions and biochemical parameters." Acta Physiologica Hungarica 95, no. 4 (2008): 395-403.

Methods:

Line 144 and 145 has a lot of problems and mistakes.

Results/ Discussion

Is it possible to separate the results and discussion section?

More comparison needed between results of this study and other similar studies in discussion section.

Health Risk Assessment:

Please note that human exposure to hazardous chemicals such as heavy metals could occur via various routes such as ingestion of contaminated soil, drinking water and inhalation of contaminated air and soil dust. Unless the daily dose calculation is based on all possible exposure pathways, the exposure dose calculation only from the dermal exposure may not be  enough. Please note this clearly as a limitation of study.

Author Response

Dear Reviewer,

On behalf of my co-authors I would like to thank you for careful reading of our manuscript and for very precise comments. Below please (and in PDF file) find the answers to your comments. The changes in the manuscript were also marked.

I hope you will find our revised manuscript suitable for publication in Sustainability.

Sincerely yours,

Nicole Nawrot

Below there are the answers for comments:

Introduction:

After line 36 talk about some of the systems it can affect in the body such as the:

1)      cardiovascular system

Lanphear, Bruce P., Stephen Rauch, Peggy Auinger, Ryan W. Allen, and Richard W. Hornung. "Low-level lead exposure and mortality in US adults: a population-based cohort study." The Lancet Public Health 3, no. 4 (2018): e177-e184.

 Obeng-Gyasi, Emmanuel, Rodrigo X. Armijos, M. Margaret Weigel, Gabriel M. Filippelli, and M. Aaron Sayegh. "Cardiovascular-Related Outcomes in US Adults Exposed to Lead." International journal of environmental research and public health 15, no. 4 (2018): 759.

2)      Renal system

 Harari, Florencia, Gerd Sallsten, Anders Christensson, Marinka Petkovic, Bo Hedblad, Niklas Forsgard, Olle Melander et al. "Blood Lead Levels and Decreased Kidney Function in a Population-Based Cohort." American Journal of Kidney Diseases (2018).

 Lin, Ja-Liang, Dan-Tzu Lin-Tan, Kuang-Hung Hsu, and Chun-Chen Yu. "Environmental lead exposure and progression of chronic renal diseases in patients without diabetes." New England Journal of Medicine 348, no. 4 (2003): 277-286.

3)      Hepatic system

 Obeng-Gyasi, Emmanuel, Rodrigo X. Armijos, M. Margaret Weigel, Gabriel Filippelli, and M. Aaron Sayegh. "Hepatobiliary-Related Outcomes in US Adults Exposed to Lead." Environments 5, no. 4 (2018): 46.

Can, S., C. Bağci, M. Ozaslan, A. I. Bozkurt, B. Cengiz, E. A. Cakmak, R. Kocabaş, E. Karadağ, and M. Tarakçioğlu. "Occupational lead exposure effect on liver functions and biochemical parameters." Acta Physiologica Hungarica 95, no. 4 (2008): 395-403.

Answer: Thank you for your recommendation of our manuscript. In L53-61 the information about heavy metals toxicity, health risk and impact on human were added. Also the literature list was complemented.

L53-61: HMs due to their toxicity, persistence and potential to bioaccumulate present a serious problem of environmental pollution [18,19]. However little is known as regards the influence of possible health risk due to HMs contamination in bottom sediments. Inhabitants of urban areas could easily be exposed to metals in the environment via three pathways: direct inhalation, ingestion and dermal contact absorption [20,21,22]. The human exposure to hazardous elements contained in sediments, soils, drinking water etc. are most often calculated as a component of three pathways. HMs occurrence can be especially detrimental to the children health; due to weakness of their immune system [23]. The particularly dangerous effects of heavy metals have been extensively reported for cardiovascular system [24,25], renal system [26,27] and hepatic system [28,29].”

Methods:

Line 144 and 145 has a lot of problems and mistakes.

Answer: Thank you very much for pointing this out. The mistakes were corrected (now it is contain in L159-162):

L159-162: “In this study, the CmBackground value was defined on the basis of average concentration of the analysed metals in the top layer of soil in the Gdańsk urban area [41] (research developed from 1991) – hereinafter referred as geochemical background – GB. The minimum, maximum and mean values are presented in Tab. S.3. The mean values were used for calculations.”

Results/ Discussion

Is it possible to separate the results and discussion section?

Answer: We have checked in "Instruction for Authors” that it is possible to separate the results and discussion section. We did hope that this could be more easy to follow by the reader.

More comparison needed between results of this study and other similar studies in discussion section.

More information and comparison were included in the manuscript (L201-202, L234-240, L253-257). We hope this contributed to improvement of its quality.

L201-202: “The pH ranges from 6.8 in Potokowa, Ogrodowa and Grunwaldzka RTs to 7.0 in Nowiec II and Chłopska RTs. The contents of organic matter were in the ranges 2-13%.”

L234-240: In comparison to research conducted by Nowell et al. [48] with regard to contamination of stream bottom sediments in relation to urbanization in the USA, lower concentrations of HMs were observed in the current study for all HMs with exception of Pb, Zn and Cu in Potokowa RT. The concentrations of Pb and Zn were on similar level in our study and in the study by Nowell et al. [48] while the Cu concentration was definitely higher in Potokowa tank. The mean values observed in Nowell et al. study sediments were as follows: Cu – 80 , Zn – 300, Pb – 100, Cd – 0.8, Cr – 80 and Ni – 50 [mg/kg d.w.].”

L253-257: „In the study conducted by Nowell et al. [48] not only the mean HM concentrations for USA bottom sediments but also the highest values were reported. The highest value of Cu concentrations in the USA was equal to 500 mg/kg d.w. (in Boston). When comparing this report to our study it is obvious that the Cu concentration measured at Orłowska IN (1114 mg/kg d.w.) is almost 2 times as high.”

Health Risk Assessment:

Please note that human exposure to hazardous chemicals such as heavy metals could occur via various routes such as ingestion of contaminated soil, drinking water and inhalation of contaminated air and soil dust. Unless the daily dose calculation is based on all possible exposure pathways, the exposure dose calculation only from the dermal exposure may not be  enough. Please note this clearly as a limitation of study.

Thank you for this comment. Generally, we have considered only the dermal pathway as we believed it to be the most likely to occur during the dredging works. As you suggested it was clearly explained in the manuscript (L353-355). “In this calculation ADD via ingestion and inhalation contact were omitted, assuming no consumption and no dust generated during dredging (due to high water content in the sediments)”. It was also mentioned that “The complex analyses of health hazard would require also evaluation of ingestion and inhalation pathways”.

The supporting information are presented in:

L171-173: “Hazard quotient HQ [21,22,34,36,37,41] is used to evaluate the health risk to humans related to sediments dredging from RTs. This kind of evaluation is used to quantify both carcinogenic and non-carcinogenic risks to humans via three exposure pathways: ingestion, dermal contact and inhalation.”

L187: “The sum of HQ is defined as hazard index and refers to the total risk of non-carcinogenic element via 3 exposure pathways.”

L191-193: “In this study the HQ was calculated basing only on the one pathway (dermal contact), due to the assumption of dermal contact with periodically dredged sediments.”

L355-356: “The complex analyses of health hazard would require also evaluation of ingestion and inhalation pathways.”

Thank you once again for the time spent on reviewing of our manuscript and for all your comments and remarks that we found very helpful during revision.

Round  2

Reviewer 3 Report

Well done on the revisions.